# Potential Application of the Myocardial Scintigraphy Agent [^123^I]BMIPP in Colon Cancer Cell Imaging

**DOI:** 10.3390/ijms25147747

**Published:** 2024-07-15

**Authors:** Kakeru Sato, Yuka Hirayama, Asuka Mizutani, Jianwei Yao, Jinya Higashino, Yuto Kamitaka, Yuka Muranaka, Kana Yamazaki, Ryuichi Nishii, Masato Kobayashi, Keiichi Kawai

**Affiliations:** 1Division of Health Sciences, Graduate School of Medical Sciences, Kanazawa University, 5-11-80 Kodatsuno, Kanazawa 920-0942, Japan; stkk0323@g.u-fukui.ac.jp (K.S.); yuka0916@stu.kanazawa-u.ac.jp (Y.H.); sudayjw@outlook.com (J.Y.); hjmr0427@icloud.com (J.H.); yuto.kamitaka419@gmail.com (Y.K.); 2Radiological Center, University of Fukui Hospital, 23-3 Matsuokashimoaizuki, Eiheiji, Fukui 910-1193, Japan; 3Faculty of Health Sciences, Institute of Medical, Pharmaceutical and Health Sciences, Kanazawa University, 5-11-80 Kodatsuno, Kanazawa 920-0942, Japan; mizutani.a@staff.kanazawa-u.ac.jp (A.M.); kei@mhs.mp.kanazawa-u.ac.jp (K.K.); 4Research Team for Neuroimaging, Tokyo Metropolitan Institute for Geriatrics and Gerontology, 35-2 Sakae-cho, Itabashi-ku, Tokyo 173-0015, Japan; 5Department of Radiological Technology, Faculty of Health Science, Juntendo University, 2-1-1 Hongo, Bunkyo-ku, Tokyo 113-8421, Japan; y.muranaka.vt@juntendo.ac.jp; 6Department of Molecular Imaging and Theranostics, Institute for Quantum Medical Science, National Institutes for Quantum Science and Technology, 4-9-1 Anagawa, Inage-ku, Chiba 263-8555, Japan; yamazaki.kana@qst.go.jp; 7Department of Integrated Health Sciences, Graduate School of Medicine, Nagoya University, 1-1-20 Daiko Minami, Higashi-ku, Nagoya 461-8673, Japan; nishii.ryuichi.g1@f.mail.nagoya-u.ac.jp; 8Biomedical Imaging Research Center, University of Fukui, 23-3 Matsuokashimoaizuki, Eiheiji, Fukui 910-1193, Japan

**Keywords:** [^123^I]BMIPP, fatty acid transporter, fatty acid metabolism, CPT1, cancer imaging

## Abstract

[^123^I]β-methyl-p-iodophenyl-pentadecanoic acid ([^123^I]BMIPP), which is used for nuclear medicine imaging of myocardial fatty acid metabolism, accumulates in cancer cells. However, the mechanism of accumulation remains unknown. Therefore, this study aimed to elucidate the accumulation and accumulation mechanism of [^123^I]BMIPP in cancer cells. We compared the accumulation of [^123^I]BMIPP in cancer cells with that of [^18^F]FDG and found that [^123^I]BMIPP was a much higher accumulation than [^18^F]FDG. The accumulation of [^123^I]BMIPP was evaluated in the presence of sulfosuccinimidyl oleate (SSO), a CD36 inhibitor, and lipofermata, a fatty acid transport protein (FATP) inhibitor, under low-temperature conditions and in the presence of etomoxir, a carnitine palmitoyl transferase I (CPT1) inhibitor. The results showed that [^123^I]BMIPP accumulation was decreased in the presence of SSO and lipofermata in H441, LS180, and DLD-1 cells, suggesting that FATPs and CD36 are involved in [^123^I]BMIPP uptake in cancer cells. [^123^I]BMIPP accumulation in all cancer cell lines was significantly decreased at 4 °C compared to that at 37 °C and increased in the presence of etomoxir in all cancer cell lines, suggesting that the accumulation of [^123^I]BMIPP in cancer cells is metabolically dependent. In a biological distribution study conducted using tumor-bearing mice transplanted with LS180 cells, [^123^I]BMIPP highly accumulated in not only LS180 cells but also normal tissues and organs (including blood and muscle). The tumor-to-intestine or large intestine ratios of [^123^I]BMIPP were similar to those of [^18^F]FDG, and the tumor-to-large-intestine ratios exceeded 1.0 during 30 min after [^123^I]BMIPP administration in the in vivo study. [^123^I]BMIPP is taken up by cancer cells via CD36 and FATP and incorporated into mitochondria via CPT1. Therefore, [^123^I]BMIPP may be useful for imaging cancers with activated fatty acid metabolism, such as colon cancer. However, the development of novel imaging radiotracers based on the chemical structure analog of [^123^I]BMIPP is needed.

## 1. Introduction

In the field of nuclear medicine, many diseases are diagnosed using positron emission tomography (PET) or single photon emission computed tomography (SPECT) or treated using radiopharmaceuticals [1,2]. PET with 2-deoxy-2-[fluorine-18]fluoro-D-glucose ([^18^F]FDG) is the most widely used technique radiotracer in nuclear medicine for cancer imaging [3,4,5]. [^18^F]FDG, a glucose analog, is taken up by cells and undergoes phosphorylation through the same mechanism as glucose. However, unlike glucose, [^18^F]FDG does not enter the glycolytic pathway and accumulates inside the cells [6]. Consequently, [^18^F]FDG reflects glucose metabolism and enables imaging of tumors that do not exhibit morphological changes [3].

Cancer cells consume more glucose and fatty acids as energy sources than normal cells [7,8]. Fatty acids are composed of hydrocarbon chains and carboxyl groups and are categorized into short-chain fatty acids (SCFAs), medium-chain fatty acids, and long-chain fatty acids (LCFAs) based on their carbon content [9]. PET imaging with ^11^C-acetate, a type of SCFA, reflects fatty acid synthesis [10]. ^11^C-acetate is taken into the cancer cell through monocarboxylate transporters [11] and converted to acetyl CoA by acetyl CoA synthase, which is then converted to LCFAs by fatty acid synthase [8,12,13]. Because the accumulation of ^11^C-acetate is correlated with the function of fatty acid synthesis enzymes, ^11^C-acetate PET imaging is used to evaluate fatty acid synthase [14].

In addition to fatty acid synthesis, fatty acid oxidation (FAO), which produces energy through β-oxidation of LCFA, is also an important source of energy for cancer cells [15]. FAO has attracted attention as a target for cancer therapy, as it has recently been shown that loading with FAO inhibitors suppresses cancer cell growth [16]. To confirm the effect of FAO inhibitors, etc., several fluorescent probes have been developed [17,18]. Fluorescence imaging has the advantage of no radiation exposure, but the sensitivity is poor, and clinical imaging in humans is difficult [19]. Therefore, nuclear medicine probes, which enable clinical imaging, would be useful. However, there are no nuclear medicine imaging agents for cancer that reflect FAO. [^123^I]β-methyl-p-iodophenyl-pentadecanoic acid ([^123^I]BMIPP), whose parent compound is palmitic acid, is currently used for nuclear medicine imaging of ischemic heart diseases as myocardial fatty acid metabolism scintigraphy [20]. In the fatty acid metabolism of [^123^I]BMIPP myocardial scintigraphy, FAO is involved in [^123^I]BMIPP accumulation. Although [^123^I]BMIPP accumulates in cancer cells in patients with cancer as well [21], the accumulation mechanism of [^123^I]BMIPP in cancer cells remains unclear. Thus, this study aimed to elucidate the accumulation mechanism of [^123^I]BMIPP in cancer cells.

## 2. Results

Figure 1 shows the accumulation of [^123^I]BMIPP and [^18^F]FDG in (a) H441, (b) PC-14, (c) LS180, and (d) DLD-1cells. In all cancer cells, the accumulation of [^123^I]BMIPP was much higher than that of [^18^F]FDG early after its administration. It was also shown that the accumulation of [^123^I]BMIPP increased over time.

Figure 2 shows the accumulation of [^123^I]BMIPP in (a) H441, (b) PC-14, (c) LS180, and (d) DLD-1 cells in the presence of SSO and lipofermata. The accumulation of [^123^I]BMIPP decreased in H441, LS180, and DLD-1 cells when loaded with SSO and lipofermata. In addition, the decrease in the accumulation of [^123^I]BMIPP was more pronounced in SSO-loaded cells than in lipofermata-loaded cells. Conversely, the accumulation of [^123^I]BMIPP in PC-14 was not significantly changed by the two inhibitor loadings.

Figure 3 shows the accumulation of [^123^I]BMIPP in (a) H441, (b) PC-14, (c) LS180, and (d) DLD-1 under low-temperature conditions (4 °C). In all cancer cells, the accumulation of [^123^I]BMIPP was considerably reduced at low temperatures compared to that of the control at 37 °C.

Figure 4 shows the accumulation of [^123^I]BMIPP in (a) H441, (b) PC-14, (c) LS180, and (d) DLD-1 cells with etomoxir, an FAO inhibitor, and loading. The accumulation of [^123^I]BMIPP was significantly increased in all cancer cells when loaded with etomoxir. The increase in the accumulation of [^123^I]BMIPP was more prominent in the lung adenocarcinoma cell lines H441 and PC-14 than in the colon cancer cell lines LS180 and DLD-1.

The biological distribution of [^123^I]BMIPP in LS180 tumor-bearing mice is shown in Table 1, and the tumor-to-blood, tumor-to-large-intestine, tumor-to-small intestine, and tumor-to-muscle ratios of [^123^I]BMIPP are shown in Table 2. The biological distribution of [^18^F]FDG in LS180 tumor-bearing mice is shown in Table 3, and the tumor-to-blood, tumor-to-large-intestine, tumor-to-small intestine, and tumor-to-muscle ratios of [^18^F]FDG are shown in Table 4. The accumulation of [^123^I]BMIPP in the blood increased up to 30 min after its administration. Conversely, the accumulation of [^18^F]FDG in the blood peaked 5 min after its administration and decreased over time. There was little accumulation of [^123^I]BMIPP in the brain and thyroid gland. At 5 min after its administration, the accumulation of [^123^I]BMIPP in the heart and liver was observed, followed by a decrease in its accumulation over time. The accumulation of [^123^I]BMIPP in the large intestine tended to increase over time. At 30 min after its administration, the accumulation of [^123^I]BMIPP in LS180 cells was at its highest (2.50% ± 0.25%ID/g). The accumulation of [^18^F]FDG in the LS180 cells similarly peaked at 30 min after its administration (3.02 ± 0.44%ID/g) but decreased considerably at 60 min after its administration (0.99 ± 0.08%ID/g). The tumor-to-blood ratio was 0.45, the tumor-to-large-intestine ratio was 1.63, and the tumor-to-muscle ratio was 1.16 at 30 min after the administration of [^123^I]BMIPP. The maximum tumor-to-large-intestine ratio was 1.96 at 5 min after [^123^I]BMIPP administration. Furthermore, the tumor-to-blood ratio was 3.02, the tumor-to-large-intestine ratio was 1.70, and the tumor-to-muscle ratio was 1.46 at 30 min after the administration of [^18^F]FDG. These results indicated [^123^I]BMIPP might be useful for colon cancer imaging.

## 3. Discussion

[^123^I]BMIPP accumulates in the cancer cells of patients with cancer [21]. However, cancer imaging with [^123^I]BMIPP has not been used as a routine clinical examination. Our in vitro studies demonstrated significantly higher [^123^I]BMIPP accumulation in all cancer cells than [^18^F]FDG (Figure 1), which was similar to [^18^F]FDG accumulation in cancer cells in Iwamoto’s study [22]. This suggests that cancer cells may take up more LCFA than glucose as a nutrient source. LS180 cells showed the highest accumulation of [^123^I]BMIPP at 5 min after its administration among all cell lines, suggesting that [^123^I]BMIPP is taken up into cancer cells via transporters. Furthermore, the two lung cancer cells showed a more pronounced increase in accumulation from 30 to 60 min compared to the two colon cancer cells, suggesting that [^123^I]BMIPP could be retained in the cells by its metabolism.

Regarding the mechanism by which [^123^I]BMIPP accumulates in cancer cells, we focused on the fatty acid transporters CD36 and FATPs because LCFAs enter cells through fatty acid transporters of CD36 and FATPs on the cell membrane [23,24,25,26], although there are other fatty acid transporters such as fatty acid binding proteins (FABPs) [27]. Notably, CD36 mediates the uptake of [^123^I]BMIPP into the myocardium [28]. Since no specific inhibitor for all subtypes of FABPs has been reported, [^123^I]BMIPP accumulation was assessed in the presence of SSO, a CD36 inhibitor, and lipofermata, a FATP inhibitor. The accumulation of [^123^I]BMIPP was reduced in the presence of SSO and lipofermata in H441, LS180, and DLD-1 cells but not in PC-14 (Figure 2). The expression of CD36 and FATPs in PC-14 cells may be lower than those in H441, LS180, and DLD-1 cells. However, FABPs may be more involved with respect to [^123^I]BMIPP uptake into PC-14 because [^123^I]BMIPP in PC-14 showed similar high accumulation compared to other cell lines. In H441, LS180, and DLD-1, the reduction was more pronounced with SSO than with lipofermata. Although it has been reported that [^123^I]BMIPP uptake in the myocardium is mediated by CD36 [28], [^123^I]BMIPP is primarily transported into cancer cells by CD36 and FATP, which is newly evident in cancer.

The accumulation and retention of [^123^I]BMIPP in the heart is dependent on fatty acid metabolism in the cytoplasm and mitochondria after myocardial uptake via CD36 and FATP1, 4, and 6 [29,30,31]. In addition, we have shown that [^123^I]BMIPP accumulates in bacteria through metabolic activity [32]. In the symbiotic theory of biology, mitochondria are thought to have originally been aerobic bacteria [33]. Thus, we investigated the metabolism of [^123^I]BMIPP in cancer cells and compared it with [^123^I]BMIPP accumulation under low-temperature conditions and 37 °C. Because the accumulation of [^123^I]BMIPP was significantly reduced at 4 °C compared to 37 °C in all cancer cell lines (Figure 3), [^123^I]BMIPP in cancer cells also accumulates depending on its metabolic activity.

LCFAs are transported into cells and subsequently incorporated into mitochondria, where they undergo β-oxidation to generate energy [15]. However, native fatty acids cannot directly penetrate mitochondrial membranes. Therefore, fatty acids are converted to acyl CoA, and the enzyme known as carnitine palmitoyl transferase I (CPT1) converts acyl CoA and carnitine to acylcarnitine, which can be transported into the mitochondria, where it is involved in energy production through β-oxidation [16,34]. Etomoxir, an inhibitor of FAO, inhibits the metabolic pathway by blocking CPT1 [16,35]. It has been shown that when loaded with etomoxir, the contribution of β-oxidation of [^123^I]BMIPP in the myocardium decreased from 10% to 0% [36]. In the myocardium, a small portion of [^123^I]BMIPP is incorporated into the mitochondria, and most of the other molecules remain in the cytoplasm (mainly triglycerides). Conversely, when cancer cells were exposed to etomoxir at a high concentration for 60 min, there was a notable increase in the accumulation of [^123^I]BMIPP compared with the control in all cancer cells (Figure 4). Herein, the increase in [^123^I]BMIPP accumulation at 60 min after etomoxir loading, which may mainly include metabolites of [^123^I]BMIPP, was higher in lung adenocarcinoma cell lines than in colon cancer cell lines (Figure 4). Reportedly, etomoxir does not alter cardiac long-chain fatty acid uptake via CD36 and FABPs [37]. Therefore, etomoxir loading did not affect the intracellular accumulation of fatty acids. Thus, etomoxir loading inhibited CPT1 function and might increase [^123^I]BMIPP accumulation into cytoplasmic triglycerides by altering cellular function, excluding transporter function in lung adenocarcinoma rather than colon cancer. Furthermore, the uptake of palmitic and oleic acids is markedly increased in CPT1 knockdown cells, which is close to the inhibition state of CPT by etomoxir loading, but the mechanism is unclear [38]. Further investigation is needed to identify how CPT1 knockdown and CPT1 inhibition by etomoxir loading in cancer cells increase fatty acid uptake using metabolic pathway analysis of fatty acids, including [^123^I]BMIPP. Based on the results of the present and Yao’s studies [38], [^123^I]BMIPP would be taken up by mitochondria via CPT1.

We also investigated whether [^123^I]BMIPP highly accumulated in tumors in vivo using tumor-bearing mice implanted with LS180 cells, which exhibited the highest accumulation of [^123^I]BMIPP in vitro in the thigh. The mice were allowed to fast for 6 h pre-experiment and injected venously with [^123^I]BMIPP and [^18^F]FDG. Ikeda et al. reported a significant reduction in triglycerides and fatty acid synthase in mice after 9 or 13 h of fasting compared with mice after 6 h of fasting [39]. Herein, we primarily investigated fatty acid metabolism in cancer. Furthermore, we focused on fatty acid metabolism in cancer, speculating that fasting for >9 h could affect lipid metabolism in mice. Therefore, in this experiment, a 6 h fasting period was chosen, which did not significantly differ from that of nonfasting mice. The accumulation of [^18^F]FDG in the blood achieved peak values at 5 min after its administration and then decreased over time (Table 3), whereas the accumulation of [^123^I]BMIPP increased up to 30 min after its administration and then decreased at 60 min (Table 1). Furthermore, the accumulation of [^123^I]BMIPP in the tumor reached a maximum of 2.50%ID/mg at 30 min after its administration, with a tumor-to-blood ratio of 0.45 (Table 2).

Fatty acids bind to serum albumin in the blood [40]. The albumin binding rate of [^123^I]BMIPP was >99%, as evaluated by the ultrafiltration method of Nishi et al. [41]. Therefore, [^123^I]BMIPP accumulation in the blood was high, resulting in a low tumor-to-blood ratio. Little accumulation of [^123^I]BMIPP in the mouse brain was observed (Table 1), although the accumulation of [^123^I]BMIPP in the human brain has not been fully confirmed [42]. Since the expression of FATP1/4 on the blood–brain barrier has been confirmed in humans [43], [^123^I]BMIPP may not have an affinity for FATP1/4. Oleic acid has an affinity for FATP1 [44], and palmitic acid, the parent compound of [^123^I]BMIPP, has an affinity for FATP4 [45]. However, the chemical structure of these fatty acids is different from that of [^123^I]BMIPP, which has a benzene and methyl group, potentially explaining why [^123^I]BMIPP does not have an affinity for FATP1/4. Additionally, FATP3 and FATP5 are primarily expressed in lung adenocarcinoma and colon cancer, respectively [33]. These transporters showed some expression levels in all cancer cell lines (Table 1). FATP2, 6, and FABPs are also expressed in cancer cells [31]. Therefore, [^123^I]BMIPP may have an affinity for CD36, FATPs (excluding FATP1/4), and FABPs as fatty acid transporters.

The accumulation of [^123^I]BMIPP in whole normal tissues (such as the heart and liver) was high early after its administration and tended to decrease over time compared to its accumulation in LS180 cells. Coburn et al. reported that systemic CD36 knockout in mice did not reduce fatty acid uptake into the liver [46], suggesting that FATPs or FABPs are involved in the uptake of [^123^I]BMIPP into the liver. CD36 in the heart and FATPs or FABPs in the liver suggest higher expression levels than CD36 and FATP expressed on the cell membrane of the tumor due to the substantial impact of transporters in the early post-administration period. Additionally, [^123^I]BMIPP, a fatty acid derivative, potentially has not much affinity for efflux drug transporters and would be gradually transferred from the liver to the intestine.

[^123^I]BMIPP accumulation increased over time, up to 60 min after in vivo administration, but decreased 60 min after in vivo administration. No correlation was observed between the in vitro and in vivo results because [^123^I]BMIPP is metabolized in cancer cells and some organs, including the liver, in vivo, whereas [^123^I]BMIPP is metabolized only in cancer cells in vitro. The tumor-to-large-intestine ratios in the colon cancer cell line LS180 exceeded 1.0 during the 30 min that followed [^123^I]BMIPP administration, reaching a maximum of 1.96 at 5 min of administration. Because it has been reported that the blood flow in the colon is lower than that in other organs [47], [^123^I]BMIPP may have resulted in the tumor-to-large-intestine ratios exceeding 1.0 even though the tumor-to-blood ratios did not exceed 1.0. The maximum tumor-to-large-intestine ratio was 1.63 at 30 min after [^18^F]FDG administration. [^123^I]BMIPP can be applied to imaging bacterial infections [32]. Herein, the ratio of infected to uninfected areas was approximately 1.3. Thus, a tumor-to-large-intestine ratio of 1.6 at 30 min after [^123^I]BMIPP administration could be used to visualize the tumor. Therefore, [^123^I]BMIPP could be used for imaging cancers with activated fatty acid metabolism, such as colon cancer. In addition, the accumulation of [^123^I]BMIPP in muscle was at the same level as that in cancer. Increased FAO has been shown to be activated in skeletal muscle tissues after exercise [48,49], and the mice used in this experiment were active before and after [^123^I]BMIPP administration, which may have increased [^123^I]BMIPP accumulation in the skeletal muscle. Because patients can also rest until the starting time of examination after [^123^I]BMIPP administration, [^123^I]BMIPP could visualize cancer cells in humans, unlike the biological distribution of mice.

Currently, the primary clinical examination methods for colorectal cancer are the fecal occult blood test (FOBT) and colonoscopy [47,50]. While FOBT is relatively easy to perform, it can give false-positive or false-negative results [51,52]. In addition, colonoscopy allows for direct observation of the inside of the intestine but imposes a significant burden on the patient because of the insertion of the endoscope into the body, along with the associated risks of bleeding and perforation of the digestive tract [53]. Therefore, nuclear medicine imaging, which is less burdensome and may enable the early detection of tumors with subtle morphological changes, can prove valuable in the diagnosis of colorectal cancer. The results of this study suggest that [^123^I]BMIPP can visualize fatty acid metabolism accumulated in mitochondria via CPT1 and is useful for early colon cancer imaging. Furthermore, loading palmitic acid (the precursor compound of [^123^I]BMIPP) onto colorectal cancer induces ferroptosis, leading to cell death [50]. Therefore, [^123^I]BMIPP imaging for the measurement of fatty acid metabolism is a valuable tool for colon cancer. However, SPECT imaging could not be performed in colon cancer-bearing mice with implants in the thigh because the tumor-to-muscle ratio was less than 1.0 (Table 3). Thus, new techniques for implanting colon cancer cells into the large intestine for SPECT imaging are needed.

[^123^I]BMIPP might be useful for colon cancer imaging. However, [^123^I]BMIPP binds to >99% of serum albumin, causing its retention in the blood, and imaging of lung cancer, liver cancer, and other types of cancer would be difficult because of the large [^123^I]BMIPP accumulation in the heart, blood, and other trunk tissues. Therefore, new imaging radiotracers based on the chemical structure analog of [^123^I]BMIPP are needed to enable fatty acid metabolism imaging of various types of cancer.

## 4. Materials and Methods

[^123^I]BMIPP was purchased from Nihon Medi-Physics Co., Ltd. (Tokyo, Japan). [^18^F]FDG was synthesized at the PET facility of Kanazawa University.

### 4.1. Using Cancer Cell Lines

The human-derived lung adenocarcinoma cancer cell lines H441 (American Type Culture Collection, Manassas, VA, USA) and PC-14 (RIKEN Cell Bank, Tsukuba, Japan), and the human-derived colon cancer cell lines LS180 and DLD-1 (American Type Culture Collection, Manassas, VA, USA) were selected in this study because lung cancer and colon cancer have the number one and number two cancer mortality rates in the world, respectively [54,55]. H441 was cultured in the RPMI-1640 medium (FUJIFILM Wako Chemical, Osaka, Japan). PC-14 and DLD-1 were cultured in Dulbecco’s Modified Eagle’s Medium (FUJIFILM Wako Chemicals, Tsukuba, Japan), and LS180 was cultured in Eagle’s Minimum Essential Medium (E-MEM, FUJIFILM Wako Chemicals, Tokyo, Japan). All cell lines were cultured at 37 °C in a 5% CO_2_ atmosphere. In addition, all culture media were mixed with 10% fetal bovine serum and 1% sodium pyruvate.

### 4.2. Accumulation of [^123^I]BMIPP and [^18^F]FDG in Cancer Cells

Each cancer cell was seeded in a 12-well plastic plate at a density of 1.0 × 10^5^ cells/well. Approximately 1 day after seeding, the cells were preincubated in phosphate-buffered saline (PBS, pH 7.4) for approximately 5 min and then incubated with [^123^I]BMIPP (37 kBq/well, 1.08 fmol/kBq) and [^18^F]FDG (370 kBq/well, 0.35 fmol/kBq) for 5, 10, 30, and 60 min at 37 °C (*n* = 4). After incubation, the cancer cells were washed twice with PBS and lysed with 0.1 M NaOH. Intracellular radioactivity was measured using a gamma counter (AccuFLEX γ7000, Hitachi Aloka Medical, Tokyo, Japan). The measurement results were expressed as the percentage injected dose (%ID)/mg protein using the amount of intracellular protein measured with a protein assay kit (Thermo Fisher Scientific, Waltham, MA, USA).

### 4.3. Accumulation of [^123^I]BMIPP in Cancer Cells Using a Fatty Acid Transporter Inhibitor

Each cancer cell was seeded in a 12-well plastic plate at a density of 1.0 × 10^5^ cells/well. Approximately 1 day after seeding, the cells were preincubated in PBS for approximately 5–10 min and then incubated with [^123^I]BMIPP (37 kBq/well, 40 fmol/37 kBq) and 1.0 mM sulfosuccinimidyl oleate (SSO, Cayman Chemical, Ann Arbor, MI, USA), an inhibitor of CD36 [56], or 1.0 mM lipofermata (Cayman Chemical, Ann Arbor, MI, USA), an inhibitor of fatty acid transport proteins (FATPs) [57], for 5 min, at which transporters are active (*n* = 4). Subsequent procedures followed the method outlined in Section 4.2.

### 4.4. Accumulation of [^123^I]BMIPP in Cancer Cells under Low-Temperature Conditions

Each cancer cell was seeded in a 12-well plastic plate at a density of 1.0 × 10^5^ cells/well. Approximately one day after seeding, the cells were preincubated in ice-cold PBS for approximately 5 min and then incubated with [^123^I]BMIPP (37 kBq/well, 1.08 fmol/kBq) for 5, 10, 30, and 60 min under ice-cold conditions at 4 °C (*n* = 4). Subsequent procedures followed the method outlined in Section 4.2.

### 4.5. Accumulation of [^123^I]BMIPP in Cancer Cells Using a Fatty Acid Oxidation Inhibitor

Each cancer cell was seeded in a 12-well plastic plate at a density of 1.0 × 10^5^ cells/well. Approximately 1 day after seeding, the cells were preincubated in PBS for approximately 5 min and then incubated with [^123^I]BMIPP (37 kBq/well, 1.08 fmol/kBq) and 1.0 mM etomoxir, an inhibitor of FAO [58], for 60 min (*n* = 4). Subsequent procedures followed the method outlined in Section 4.2.

### 4.6. Biological Distribution of [^123^I]BMIPP in Tumor-Bearing Mice

All animal experiments conducted in this study were performed in compliance with the ethical standards of Kanazawa University (Animal Care Committee of Kanazawa University, AP-224349), international standards for animal welfare, and institutional guidelines. LS180 was adjusted to a concentration of 5.0 × 10^6^ cells/100 µL using serum-free E-MEM and transplanted under the skin of the right thigh of BALB/c male mice (4 weeks old, SLC Inc., Hamamatsu, Japan). Tumor-bearing mice (with a cancer diameter of 0.5–1.0 cm) approximately 2 weeks after transplantation were used in the experiments. The mice were made to fast for 6 h before the experiment and then injected with [^123^I]BMIPP (37 kBq/mouse, 1.08 fmol/kBq) and [^18^F]FDG (370 kBq/mouse, 0.35 fmol/kBq) through the tail vein. After 5, 10, 30, and 60 min of [^123^I]BMIPP administration (*n* = 3), blood samples were drawn from the hearts of the mice under isoflurane anesthesia (FUJIFILM Wako Pure Chemical Corporation, Osaka, Japan), and the mice were euthanized by cervical dislocation. Several organs (the brain, thyroid, heart, lungs, pancreas, liver, stomach, spleen, kidney, small intestine, large intestine, and muscle) and the tumor were collected, and their radioactivity was measured using a gamma counter. The results for the brain, heart, lungs, pancreas, liver, spleen, kidney, small intestine, large intestine, muscle, and tumor were normalized to the weight of the respective organs using an electronic balance (AUX220, Shimadzu Corporation, Kyoto, Japan) and expressed as %ID/g. The results for the thyroid, stomach, small intestine, and large intestine are expressed as %ID, and the large intestine is expressed as both. The intestine was averaged for the small and large intestines.

### 4.7. Statistical Analysis

The *p*-values were calculated using a two-tailed paired Student’s *t*-test for comparisons between the two groups using GraphPad Prism 8 statistical software (GraphPad Software, Inc., La Jolla, CA, USA). A *p*-value of 0.01 or 0.05 was considered statistically significant.

## 5. Conclusions

[^123^I]BMIPP accumulates highly in cancer cells compared to [^18^F]FDG in vitro, and the colon cancer-to-large-intestine ratio exceeds 1.0 during the 30 min that follows [^123^I]BMIPP administration in vivo. As the accumulation mechanism, [^123^I]BMIPP is mainly taken up by cancer cells via the fatty acid transporters CD36 and FATPs and incorporated into mitochondria via CPT1. Therefore, [^123^I]BMIPP may be useful for imaging cancers with activated fatty acid metabolism, such as colon cancer. However, novel imaging radiotracers based on the chemical structure analog of [^123^I]BMIPP are needed.

## Figures and Tables

**Figure 1 ijms-25-07747-f001:**
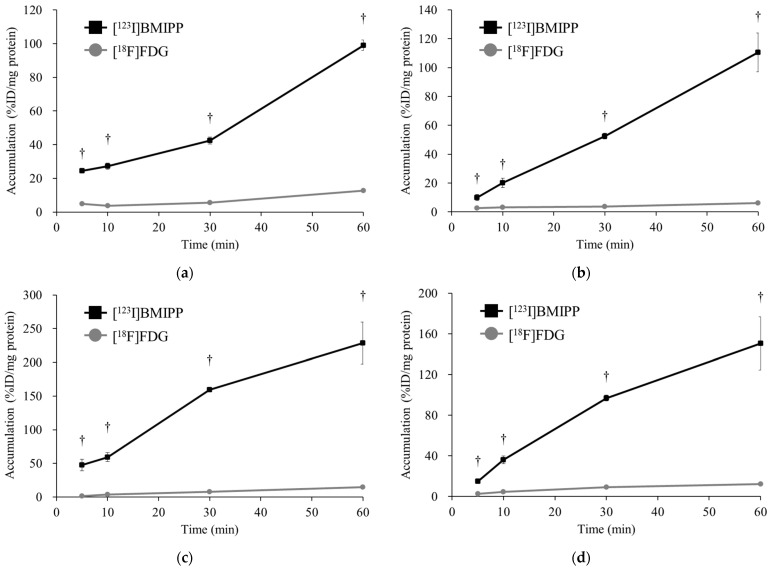
The accumulation of [^123^I]BMIPP and [^18^F]FDG in (**a**) H441, (**b**) PC-14, (**c**) LS180, and (**d**) DLD-1 at 5, 10, 30, and 60 min after the respective injections (*n* = 4). All cancer cell lines took up [^123^I]BMIPP much more than [^18^F]FDG early after its administration. The accumulation of [^123^I]BMIPP in LS180 is the highest among the four cell lines. This result suggested that cancer cells might uptake more LCFA than glucose. ^†^
*p* < 0.01 and vs. [^18^F]FDG.

**Figure 2 ijms-25-07747-f002:**
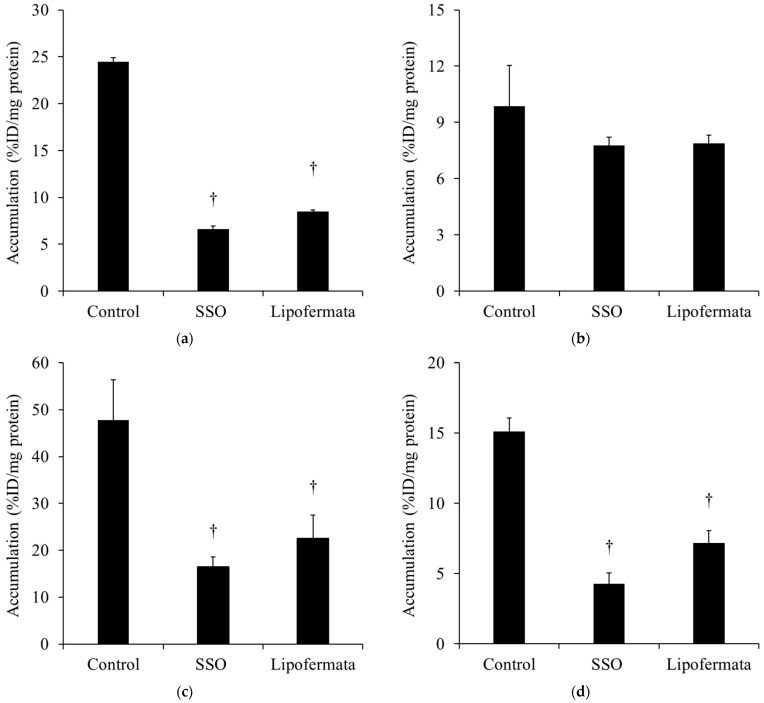
The accumulation of [^123^I]BMIPP in (**a**) H441, (**b**) PC-14, (**c**) LS180, and (**d**) DLD-1 under SSO, an inhibitor of CD36, or lipofermata, an inhibitor of FATP (*n* = 4). The accumulation of [^123^I]BMIPP was significantly decreased in H441, LS180, and DLD-1 when [^123^I]BMIPP was administered at the same time as SSO or lipofermata and taken up 5 min after. Alternatively, there was no significant difference in the accumulation of [^123^I]BMIPP in PC-14 when the inhibitor was loaded. These results indicate that [^123^I]BMIPP is transported into cancer cells via CD36 and FATP, with CD36 contributing more than FATP. ^†^
*p* < 0.01 and vs. control.

**Figure 3 ijms-25-07747-f003:**
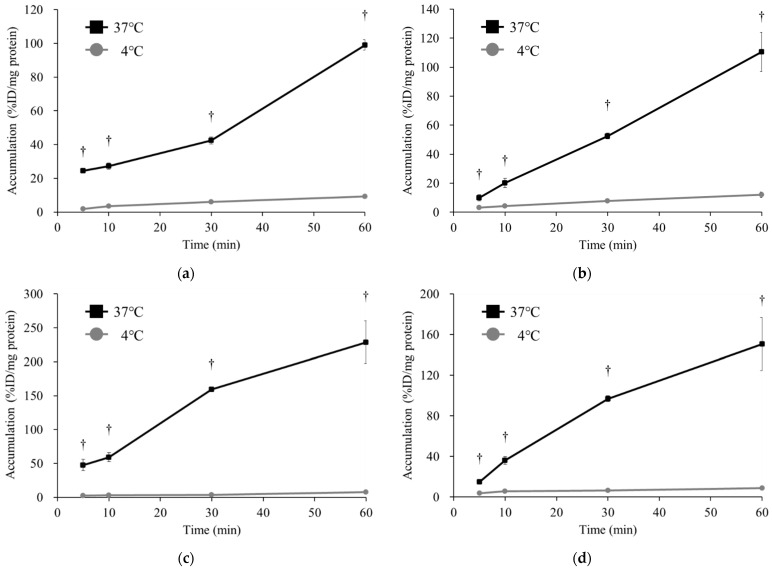
The accumulation of [^123^I]BMIPP in (**a**) H441, (**b**) PC-14, (**c**) LS180, and (**d**) DLD-1 at 37 °C or 4 °C at 5, 10, 30, and 60 min (*n* = 4). In low-temperature conditions, the accumulation of [^123^I]BMIPP was greatly decreased in all cell lines. Hence, [^123^I]BMIPP also accumulates in cancer cells depending on its metabolic activity. ^†^
*p* < 0.01 and vs. 4 °C.

**Figure 4 ijms-25-07747-f004:**
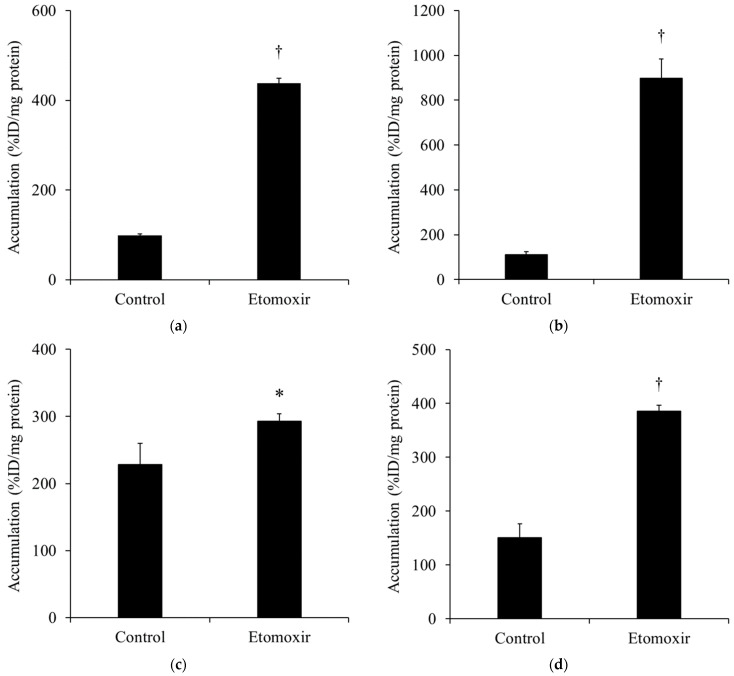
The accumulation of [^123^I]BMIPP in (**a**) H441, (**b**) PC-14, (**c**) LS180, and (**d**) DLD-1 under etomoxir, an inhibitor of fatty acid oxidation (*n* = 4). The accumulation of [^123^I]BMIPP significantly increased in all cell lines when [^123^I]BMIPP was administered at the same time as etomoxir and taken up 60 min after. The accumulation of [^123^I]BMIPP in H441 and PC-14 was greater than that in LS180 and DLD-1. Especially, the accumulation of [^123^I]BMIPP in PC-14 after etomoxir loading was more than 8 times higher compared to the control. This result suggested that the accumulation of [^123^I]BMIPP in cancer cells reflected FAO. ^†^
*p* < 0.01 and * *p* < 0.05 vs. control.

**Table 1 ijms-25-07747-t001:** Biological distribution of [^123^I]BMIPP in LS180 tumor-bearing mice.

	Time after [^123^I]BMIPP Administration			
Organ	5 min	10 min	30 min	60 min
Blood	1.99 ± 0.36	6.13 ± 1.08	5.58 ± 0.13	3.28 ± 0.60
Brain	0.06 ± 0.07	0.28 ± 0.03	0.20 ± 0.04	0.20 ± 0.05
Thyroid *	0.06 ± 0.00	0.11 ± 0.04	0.07 ± 0.03	0.09 ± 0.00
Heart	20.4 ± 0.39	15.9 ± 6.60	13.0 ± 0.97	9.15 ± 1.57
Lung	4.46 ± 0.55	7.70 ± 0.85	4.96 ± 0.43	4.50 ± 1.15
Pancreas	0.82 ± 0.13	2.72 ± 0.59	2.59 ± 0.19	1.55 ± 0.79
Liver	9.21 ± 2.11	8.63 ± 2.77	7.28 ± 0.28	3.40 ± 0.79
Stomach *	0.30 ± 0.03	0.49 ± 0.02	0.58 ± 0.08	0.41 ± 0.10
Spleen	2.92 ± 1.26	3.43 ± 1.02	2.35 ± 0.14	1.61 ± 0.62
Kidney	3.32 ± 0.86	6.04 ± 1.01	5.67 ± 0.25	3.41 ± 1.22
Large intestine *	0.54 ± 0.08	0.66 ± 0.09	0.61 ± 0.19	1.48 ± 0.15
Large intestine	0.92 ± 0.27	1.83 ± 0.10	1.53 ± 0.22	3.24 ± 0.37
Intestine *	0.67 ± 0.20	1.32 ± 0.29	1.07 ± 0.16	1.53 ± 0.11
Intestine	1.09 ± 0.28	1.96 ± 0.40	1.96 ± 0.25	2.74 ± 0.48
Muscle	2.05 ± 0.14	2.03 ± 0.64	2.18 ± 0.11	2.28 ± 0.61
Tumor	1.79 ± 0.47	1.95 ± 0.50	2.50 ± 0.25	2.40 ± 0.71

* %ID indicates percent injected dose for thyroid, stomach, small intestine, and large intestine. %ID/g indicates percent injected dose per gram of blood, brain, heart, lung, pancreas, liver, spleen, kidney, large intestine, muscle, and tumor. Values are the mean ± standard deviation obtained from four mice. The intestine was averaged for the small and large intestines (*n* = 3).

**Table 2 ijms-25-07747-t002:** Tumor-to-blood, tumor-to-large-intestine, and tumor-to-muscle ratios in LS180-bearing mice at 5, 10, 30, and 60 min after [^123^I]BMIPP injection.

	5 min	10 min	30 min	60 min
Tumor/blood	0.90	0.42	0.45	0.73
Tumor/large intestine	1.96	1.06	1.63	0.74
Tumor/intestine	1.64	0.99	1.28	0.88
Tumor/muscle	0.87	0.96	1.16	1.05

**Table 3 ijms-25-07747-t003:** Biological distribution of [^18^F]FDG in LS180 tumor-bearing mice.

	Time after [^18^F]FDG Administration			
Organ	5 min	10 min	30 min	60 min
Blood	3.58 ± 1.96	2.81 ± 0.70	1.00 ± 0.02	0.48 ± 0.09
Brain	7.41 ± 0.33	7.30 ± 0.03	7.18 ± 0.63	5.67 ± 0.88
Thyroid *	0.06 ± 0.04	0.09 ± 0.03	0.10 ± 0.04	0.08 ± 0.04
Heart	7.66 ± 1.45	6.07 ± 0.95	3.29 ± 0.25	3.55 ± 0.07
Lung	4.37 ± 0.90	2.79 ± 0.43	2.01 ± 0.05	2.81 ± 0.28
Pancreas	1.82 ± 0.15	1.86 ± 0.31	1.70 ± 0.15	1.29 ± 0.21
Liver	9.21 ± 2.11	8.63 ± 2.77	7.28 ± 0.28	3.40 ± 0.79
Stomach *	0.64 ± 0.00	0.42 ± 0.06	0.28 ± 0.06	0.27 ± 0.00
Spleen	2.54 ± 0.08	1.36 ± 0.34	1.76 ± 0.09	1.88 ± 0.25
Kidney	4.61 ± 1.87	3.79 ± 0.72	1.70 ± 0.03	1.22 ± 0.04
Large intestine *	0.49 ± 0.31	0.58 ± 0.19	0.61 ± 0.22	0.78 ± 0.43
Large intestine	2.40 ± 0.33	1.35 ± 0.42	1.78 ± 0.20	1.66 ± 0.26
Intestine *	1.61 ± 0.46	1.20 ± 0.28	0.98 ± 0.15	1.17 ± 0.43
Intestine	2.95 ± 0.63	1.68 ± 0.59	1.77 ± 0.15	1.52 ± 0.19
Muscle	2.41 ± 0.63	1.03 ± 0.03	2.06 ± 0.23	2.30 ± 0.39
Tumor	2.97 ± 0.28	1.75 ± 0.78	3.02 ± 0.44	0.99 ± 0.08

* %ID indicates percent injected dose for thyroid, stomach, small intestine, and large intestine. %ID/g indicates percent injected dose per gram of blood, brain, heart, lung, pancreas, liver, spleen, kidney, large intestine, muscle, and tumor. Values are the mean ± standard deviation obtained from four mice. The intestine was averaged for the small and large intestines (*n* = 3).

**Table 4 ijms-25-07747-t004:** Tumor-to-blood, tumor-to-large-intestine, and tumor-to-muscle ratios in LS180-bearing mice at 5, 10, 30, and 60 min after [^18^F]FDG injection.

	5 min	10 min	30 min	60 min
Tumor/blood	0.83	0.62	3.02	2.07
Tumor/large intestine	1.24	1.30	1.70	0.60
Tumor/intestine	1.01	1.04	1.70	0.65
Tumor/muscle	1.23	1.70	1.46	0.43

## Data Availability

All data are available in the article.

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
