# Peer review of "Potential Application of the Myocardial Scintigraphy Agent [123I]BMIPP in Colon Cancer Cell Imaging"

_ijms, 2024, doi:10.3390/ijms25147747_

Round 1

Reviewer 1 Report

Comments and Suggestions for Authors

Kaketu et al. submitted the manuscript entitled: Accumulation and mechanism of 123I-BMIPP, a myocardial fatty acid metabolism scintigraphy agent, in cancer cells, in which the authors tried to elucidate the mechanism of 123-BMIPP probe absorption in various cancer cells. The authors used CD36i, FATPi and CPTi to determine whether these 2 proteins are functional in probe absorption. Generally, this is an informative work and this topic will be of interest to potential readers of IJMS.

My comments are as follows.

1. The authors are suggested to include more information as research background. For example (but not limit to them), highlight the advantages of 123I-BMIPP compared with other tracers in cancer diagnosis; common use of this tracer in obesity and where the authors chose to use it to monitor tumor.

2. Including the tracer distribution images will be helpful for readers to get a direct and clear understanding about 123I-BMIPP distributions in main organs.

Author Response

  1. The authors are suggested to include more information as research background. For example (but not limit to them), highlight the advantages of 123I-BMIPP compared with other tracers in cancer diagnosis; common use of this tracer in obesity and where the authors chose to use it to monitor tumor.

Answer: Thank you for your suggestion. FAO has attracted attention as a target for cancer therapy, as it has recently been shown that loading with FAO inhibitors suppresses cancer cell growth (Clin Sci (Lond) 2019. 133(15), 1745-1758). To confirm the effect of FAO inhibitors etc., several fluorescent probes have been developed (Nat Commun. 2022. 13(1), 2533, Chem Commun (Camb). 2020;56:3023-3026.). Fluorescence imaging has the advantage of no radiation exposure, but the sensitivity is poor and the clinical imaging in human is difficult [Eur J Nucl Med Mol Imaging. 2013. 40(8), 1283-91). Therefore, nuclear medicine probes which enable the clinical imaging would be useful.

  1. Including the tracer distribution images will be helpful for readers to get a direct and clear understanding about 123I-BMIPP distributions in main organs.

Answer: Thank you for providing this insight. We thought it would be desirable to attach tumor-bearing mouse to each image. Although we believe it is possible that colon cancer can be imaged using [123I]BMIPP since the tumor-to-large intestine ratio is greater than 1.0, we transplant LS180 cells under the skin of the right thigh because we don’t have technique to implant tumor into large intestine. In table 2, we guessed that SPECT imaging in tumor bearing mice with thigh implants was so tough because tumor-to-muscle was less than 1.0. Hence, we selected only biological distribution of tumor-bearing mice and did not image the mice using SPECT.

Reviewer 2 Report

Comments and Suggestions for Authors

Thank you for an interesting article exploring the uptake of [123I] BMIPP in cancer cells.

I have no qualms recommending this article for publication. However, I found a few things which I think could improve you article, I hope you agree.

a) Please follow:

Consensus nomenclature rules for radiopharmaceutical chemistry — Setting the record straight

Heinz H. Coenen ,  Antony D. Gee, Michael Adam, Gunnar Antoni, Cathy S. Cutler, Yasuhisa Fujibayashi, Jae Min Jeong, Robert H. Mach, Thomas L. Mindt, Victor W. Pike, Albert D. Windhorst

https://www.sciencedirect.com/science/article/pii/S0969805117303189?via%3Dihub

Use square brackets i.e. [123I], [18F] throughout the article

b) Insert a drawing showing [123I] BMIPP uptake into cancer cells (so one can se haw to inhibit the up take), incorporated into mitochondria and oxidation. That why you can also show what you are blokking

c) I don’t understand how the contents of [123I] BMIPP in the blood increased up to 30 min after an injected into tail vein (Table 1) Are you sure the injection was made into a vein? Please explain.

d) Table 3:

Tumor: 2.97 ± 0.28 (5 min); 1.75 ± 0.78 (10min), 3.02 ± 0.44(30 min)  0.99 ± 0.08 (60 min)

Why is there a temporary drop at 10 min?

e) Please, write a bit about why you choose to include the 4 different cells, H441, PC-14, LS180, and  DLD-1. Why them? Why 4?

Minor things:

Page 1:

1) [123I]-β-methyl-p-iodophenyl-pentadecanoic acid ([123I]-BMIPP), which is used for nuclear medicine imaging of myocardial fatty acid metabolism, accumulates in cancer cells. However, The mechanism of accumulation remain is unknown

2) Methods: We compared the accumulation of [123I]-BMIPP in cancer cells with that of [18F]-FDG and found that [123I]-BMIPP was has a much higher accumulation than [18F]-FDG.

Page 2

3) Suggest: In the field of nuclear medicine, many diseases are diagnosed using positron emission tomography (PET) or single photon emission computed tomography (SPECT), or treated using radiopharmaceuticals [1,2].

Instead of: In the field of nuclear medicine, including positron emission tomography (PET) and single photon emission computed tomography, various diseases are diagnosed and treated using radiopharmaceuticals [1,2].

4)      is the most widely used technique radiotracer in nuclear medicine for cancer imaging

Comments on the Quality of English Language

.

Author Response

  1. a) Please follow:

Consensus nomenclature rules for radiopharmaceutical chemistry — Setting the record straight

Heinz H. Coenen ,  Antony D. Gee, Michael Adam, Gunnar Antoni, Cathy S. Cutler, Yasuhisa Fujibayashi, Jae Min Jeong, Robert H. Mach, Thomas L. Mindt, Victor W. Pike, Albert D. Windhorst

https://www.sciencedirect.com/science/article/pii/S0969805117303189?via%3Dihub

Use square brackets i.e. [123I], [18F] throughout the article

Answer: Thank you for providing these insights. We have converted all. I will be careful from now on.

  1. b) Insert a drawing showing [123I] BMIPP uptake into cancer cells (so one can se haw to inhibit the uptake), incorporated into mitochondria and oxidation. That why you can also show what you are blokking

Answer: A graphical abstract is attached that shows the contents of this study.

  1. c) I don’t understand how the contents of [123I] BMIPP in the blood increased up to 30 min after an injected into tail vein (Table 1) Are you sure the injection was made into a vein? Please explain.

Answer: Basically, the longer the time after administration, the more it is expected to be eliminated from the blood and the lower the amount of tracer in the blood. However, [123I]BMIPP is a derivative of long chain fatty acids and is known to bind to albumin in the blood. We think that this is the reason for the retention of [123I]BMIPP in the blood and the reason for the increased accumulation up to 10 minutes after administration. In fact, when our group conducted distribution experiments of [123I]BMIPP in the body of a mouse model of bacterial infection in the past, it was found that accumulation in the blood was higher at 60 minutes than at 15 minutes after administration (Pharmaceutics. 2022;14(5), 1008). This is a characteristic of long chain fatty acids, and future development of nuclear medicine imaging agents that reflect fatty acid metabolism and have low binding to albumin is needed.

  1. d) Table 3:

Tumor: 2.97 ± 0.28 (5 min); 1.75 ± 0.78 (10min), 3.02 ± 0.44(30 min), 0.99 ± 0.08 (60 min)

Why is there a temporary drop at 10 min?

Answer: Generally, [18F]FDG accumulation in cancer increases over time. However, the accumulation of [18F]FDG decreased at 10 minutes after administration. This result was attributed to measurement error, as the SD of [18F]FDG accumulation at 10 minutes after administration was as large as 0.78.

  1. e) Please, write a bit about why you choose to include the 4 different cells, H441, PC-14, LS180, and DLD-1. Why them? Why 4?

Answer: Lung cancer is the leading cause of cancer mortality in the world, and early detection is desirable (Nat Rev Clin Oncol. 2023 Sep;20(9):624-639). Following lung cancer, colon cancer is the second leading cause of cancer mortality, and its early detection is also desired (Chem Biol Interact. 2022, 368, 110170). In this study, we selected two cell lines from lung cancer and colon cancer, each of which is owned by our laboratory, and conducted the experiment.

Minor things:

Page 1:

1) [123I]-β-methyl-p-iodophenyl-pentadecanoic acid ([123I]-BMIPP), which is used for nuclear medicine imaging of myocardial fatty acid metabolism, accumulates in cancer cells. However, the mechanism of accumulation remain is unknown

Answer: Thanks for check the details. We fixed.

2) Methods: We compared the accumulation of [123I]-BMIPP in cancer cells with that of [18F]-FDG and found that [123I]-BMIPP was has a much higher accumulation than [18F]-FDG.

Methods: We compared the accumulation of 123I-BMIPP in cancer cells with that of 18F-FDG and found that 123I-BMIPP was much higher than 18F-FDG.

Page 2

Answer: We changed as following; “[123I]BMIPP was a much higher accumulation than [18F]FDG”.

3) Suggest: In the field of nuclear medicine, many diseases are diagnosed using positron emission tomography (PET) or single photon emission computed tomography (SPECT), or treated using radiopharmaceuticals [1,2].

Instead of: In the field of nuclear medicine, including positron emission tomography (PET) and single photon emission computed tomography, various diseases are diagnosed and treated using radiopharmaceuticals [1,2].

Answer: Thank you for suggestion. We fixed.

4) is the most widely used technique radiotracer in nuclear medicine for cancer imaging

Answer: We fixed.
